# Assessing the Intergenerational Linkage between Short Maternal Stature and Under-Five Stunting and Wasting in Bangladesh

**DOI:** 10.3390/nu11081818

**Published:** 2019-08-07

**Authors:** Wajiha Khatun, Sabrina Rasheed, Ashraful Alam, Tanvir M. Huda, Michael J. Dibley

**Affiliations:** 1Sydney School of Public Health, Edward Ford Building (A27), University of Sydney, Sydney, NSW 2006, Australia; 2International Centre for Diarrhoeal Disease Research Bangladesh, Mohakhali, Dhaka 1212, Bangladesh

**Keywords:** maternal stature, under-five children, stunting, wasting

## Abstract

Short maternal stature is identified as a strong predictor of offspring undernutrition in low and middle-income countries. However, there is limited information to confirm an intergenerational link between maternal and under-five undernutrition in Bangladesh. Therefore, this study aimed to assess the association between short maternal stature and offspring stunting and wasting in Bangladesh. For analysis, this study pooled the data from four rounds of Bangladesh Demographic and Health Surveys (BDHS) 2004, 2007, 2011, and 2014 that included about 28,123 singleton children aged 0–59 months born to mothers aged 15–49 years. Data on sociodemographic factors, birth history, and anthropometry were analyzed using STATA 14.2 to perform a multivariable model using ‘Modified Poisson Regression’ with step-wise backward elimination procedures. In an adjusted model, every 1 cm increase in maternal height significantly reduced the risk of stunting (relative risks (RR) = 0.960; 95% confidence interval (CI): 0.957, 0.962) and wasting (RR = 0.986; 95% CI: 0.980, 0.992). The children of the short statured mothers (<145 cm) had about two times greater risk of stunting and three times the risk of severe stunting, 1.28 times the risk of wasting, and 1.43 times the risk of severe wasting (RR = 1.43; 95% CI: 1.11, 1.83) than the tall mothers (≥155 cm). These findings confirmed a robust intergenerational linkage between short maternal stature and offspring stunting and wasting in Bangladesh.

## 1. Introduction

Undernutrition remains highly prevalent in most low- and middle-income countries (LMICs), especially in countries from South Asia. In 2016, globally among children under five years of age 39%, or 155 million, were stunted and 50%, or 52 million, were wasted [1]. In Bangladesh in 2014, the prevalence of stunting and wasting among children under five years was estimated at 36% and 14%, respectively [2]. Maternal undernutrition is a significant contributor to child undernutrition in LMICs [3]. A recent study using a pooled analysis of data from 137 developing countries showed that, in 2011, 14.4% of stunting among 44.1 million children aged under two years (6.4 million cases) was attributable to maternal undernutrition [4].

Maternal height is an indicator of intergenerational linkages between maternal and child nutrition and health [5]. As human height is inherited from parents, genetic factors should mainly determine the relationship between maternal height and offspring growth [6]. However, other factors such as metabolic programming, epigenetics, and the intergenerational transmission of poverty also play important roles [7]. Earlier research indicates that short stature of the mother is a reflection of the genetic and environmental factors, such as nutritional stresses, she experienced throughout her life course, especially at the early stage of her life [8]. Short mothers with inadequate health are less likely to be able to provide adequate nutrition to the fetus during pregnancy, resulting in small-for-gestational-age (SGA) infants. Evidence suggests that short mothers, especially those who were SGA at birth, are at higher risk of giving birth to an SGA child [9,10]. Both term and preterm SGA infants are at higher risk of being stunted and wasted during their early years [11]. A recent study that pooled Demographic and Health Survey (DHS) data from 54 LMICs reported a significant inverse association between maternal height and child undernutrition. The researchers reported that, for the offspring of short mothers (<145 cm), the risk of stunting was two times higher and the risk of wasting was 1.2 times higher, compared to those of the tallest mothers (>160 cm) [12]. Another study in India has reported similar findings, where short maternal height was associated with child undernutrition [13].

In Bangladesh data on the link between short maternal height and child malnutrition comes from a few small-scale studies [14,15]. As the data used in these studies was from restricted geographic areas and did not represent the country it was hard to generalize the findings to the whole nation. In this paper, therefore, we examined the association between maternal height and child stunting and wasting adjusting for other maternal, child, and sociodemographic covariates using a nationally representative sample. The findings of this study will help to understand the intergenerational effect of malnutrition on child anthropometry, which would benefit programs and policies aimed at reducing the prevalence of stunting and wasting among children under-five in Bangladesh.

## 2. Materials and Methods

### 2.1. Data Sources and Sampling Design

In this analysis, we pooled the data from four rounds of the Bangladesh Demographic and Health Surveys (BDHS) from 2004, 2007, 2011, and 2014 [16]. BDHS comprised nationally representative samples from ever-married women aged 15–49 years and their children with data collected on socio-demographic factors, birth history, health, and nutrition outcomes [17]. The detailed methods of the DHS survey have been described elsewhere [18]. Briefly, data were collected through interviews with the women using pre-structured questionnaires. The survey recorded the complete birth history for all the live births [19]. Also, trained interviewers measured the weight of the children under-five using SECA digital scales with a precision of 100 g, and height/length of the women and their children under-five using standard wooden boards calibrated in millimeters (mm) [20]. They measured the height for the children aged 2–5 years and recumbent length for the children aged under two years or who were shorter than 85 centimeters (cm) in height.

We used information from 28,123 live children of the mothers aged 15–49 years who were born as singleton livebirths between 1999 and 2014. In the final analysis, we excluded 4062 children (14%) with missing observations for any covariates or any biologically implausible outliers according to the World Health Organization (WHO) cut-offs for anthropometric indices (height-for-age Z-scores—HAZ, and weight-for-age Z-scores—WAZ) [20]. Our final analytical sample size was 25,635 children. We applied sampling weights to compensate for the cluster sampling design.

### 2.2. Ethics

BDHS obtained informed verbal consent from every respondent. These surveys were approved by the institutional review board of ICF Macro in Calverton, MD, USA.

### 2.3. Conceptual Framework

This study used a conceptual framework to analyze the association between maternal stature and offspring stunting and wasting along with the selected possible predictors of child undernutrition in Bangladesh (Appendix A). This framework was adapted from the 2013 UNICEF conceptual framework for determinants of child undernutrition [21] based on available data from BDHS 2004, 2007, 2011, and 2014.

### 2.4. Primary Outcomes

The primary outcomes of this study were stunting and wasting. Stunting and wasting were measured using two standard indices of physical growth of the children based on recommended guidelines of the World Health Organizations child growth standards (WHO 2006) [22]. The indices to measure stunting and wasting were height-for-age Z-score and weight-for-height Z-score, respectively. A child’s height-for-age was calculated by dividing his/her height by the median height for a child considering his/her age and sex. Similarly, a child’s weight-for-height was calculated by dividing his/her weight by the median weight for a child of that height and sex. Then, every computed number was standardised as a Z-score with a mean of 0 and a standard deviation (SD) of 1. A child was considered stunted or severely stunted if the height-for-age Z-score was less than 2 SD or less than 3 SD below the World Health Organizations determined mean scores for height-for-age. Similarly, a child was considered as wasted or severely wasted if the weight-for-height Z-score was less than 2 SD or less than 3 SD below the World Health Organizations determined mean scores for weight-for-height.

### 2.5. Main Exposure

The main exposure of this study was the maternal stature measured as height in cm. In this analysis, maternal height was used as both a continuous and a categorical variable. Continuous maternal height was used to measure the effect of every 1-cm increase of maternal height on the primary outcome (offspring stunting and wasting), while maternal height was categorized to assess the effect of the shortest maternal height category on offspring stunting and wasting compared to the tallest maternal height category. Therefore, this study categorized maternal height into four categories as follows: <145.0 cm (short stature), 149.9–145.0 cm, 154.9–150.0 cm, and ≥155.0 cm (tall stature, reference group). These catagories of maternal height were adapted from similar earlier studies [12,13,23].

### 2.6. Covariates

This study used covariates that are well-known risk factors for child undernutrition [24]. In this analysis, eleven selected covariates were selected as maternal, child, household, and community level variables. The maternal covariates were woman’s age at birth, her education, and her occupation. The child covariates were child age, sex, birth order, and birth interval. The household and community level covariates were household wealth, husband’s education, the location (rural or urban), and region of residence. Year of survey and recall period were also included as covariates. The variable “recall period” was calculated as the difference from the date of the interview to the child’s date of birth. A composite household wealth index was created from a list of household assets and facilities using the principal component analysis (PCA) to weight the individual items [25]. The household wealth index was calculated as the sum of the weighted scores and divided into quintiles for categorical analyses.

### 2.7. Statistical Analysis

Data analysis was performed using STATA 14.1 (Stata-Corp, College Station, TX, USA). For survey data analysis, command ‘svy’ were employed to adjust for the cluster sampling design in frequencies and cross-tabulations. This study applied multivariable models using Poisson Regression with robust error variance and sampling weights [26] to estimate the association between maternal height and child anthropometry adjusted for other covariates. In this method, the associations were expressed as relative risks (RR) with 95% confidence intervals (CIs). Two-tailed Wald tests were performed for measuring the level of significance, which was calculated as *p* < 0.05 or the exact *p* value when *p* < 0.001.

There were several steps in the model building process. First, an unadjusted univariable analysis was performed for the each of the covariates with the outcomes, and the covariate was selected as a factor. All the covariates that were statistically significant at the level of significance *p* < 0.25 were used to create the baseline model. Then the collinearity among the covariates was checked using the Stata command ‘collin’. This analysis showed a collinearity between birth order and birth interval. To minimize the collinearity, it was decided to exclude birth interval and keep birth order in the model, as birth order is a significant predictor of stunting among Bangladeshi children [27]. With the base model, a multi-stage backward elimination modeling technique was used to get the final model for assessing the significant factors for the primary outcome. In this process, the covariates that were not confounders, or not statistically significant at the level of significance *p* < 0.05, were excluded from the base model. In the final model, the assumption of linearity with fractional polynomial was checked for the continuous variables to choose the appropriate parameterization [28]. Moreover, the final model was checked for any interaction between main exposure and all significant covariates. The level of significance was considered at *p* < 0.01 for the interactions. All the possible interactions were considered in the final model and the non-significant interactions were gradually excluded using a backward elimination method. The significant interactions were added as effect modifiers in the full model. Then, the final models with both the factors and effect modifiers were tested for goodness-of-fit using both Pearson and deviance chi-square statistics. If these tests were not significant (*p* > 0.05), the model was considered as the best fit for Poisson regression [29].

## 3. Results

### 3.1. Prevelance of Stunting and Wasting among the Study Participants and Their Characteristics

Table 1 presents the percentage of stunted, wasted, severely stunted, and severely wasted children under-five and their maternal, birth, and socio-demographic characteristics. In this study, we analyzed data from a total of 25,635 children under-five, among whom there were 10,701 (42.1%) stunted children, 3902 (15.4%) wasted children, 4091 (15.8%) severely stunted children, and 866 (3.4%) severely wasted children. One in five stunted children and one in four severely stunted children had short mothers. About two-thirds of the mothers had at least a primary level of education. One-third of the children were their mothers’ first child. Most of the children resided in rural areas. The distribution patterns were similar across all the categories of maternal, child, and household covariates among the stunted, severely stunted, wasted, and severely wasted children (Table 1). The distributions of these covariates were also comparable across the maternal height categories among all children under-five (Appendix A).

### 3.2. Average Maternal Stature

Table 2 presents the mean maternal height in cm among the children under-five who were, or were not, stunted, wasted, severely stunted, or severely wasted. Among the 25,635 children under five included in the analysis, the mean height of the mothers was 150.8 cm (95% CI: 150.7, 150.9 cm). However, the average height among the mothers of stunted and severely stunted children was 149.3 cm (95% CI: 149.2, 149.4 cm) and 148.4 cm (95% CI: 148.3, 148.6 cm), respectively. The mean difference in maternal height was 2.6 cm (95% CI, 2.5, 2.8 cm, *p* < 0.001) lower among stunted than non-stunted children. There was also a significant difference in maternal height between severely and non-severely stunted children (Mean difference in height: 2.8 cm; 95% CI: 2.6, 3.0 cm, *p* < 0.001). The average height among the mothers of wasted children was 150.3 cm, (95% CI: 150.2, 150.5 cm), and 150.3 cm for severely wasted, (95% CI: 149.9, 150.7 cm).

### 3.3. Association of Maternal Stature with Offspring Stunting and Severe Stunting

Our results showed that both stunting and severe stunting were significantly associated with maternal, child, and household and community level covariates (Table 3, Appendix A). The adjusted models estimated that a one-centimeter increase of maternal height was a protective factor for child stunting (RR = 0.960; 95% CI: 0.957, 0.962) and severe stunting (RR = 0.941; 95% CI: 0.935, 0.946). Compared to the children of the tall mothers (≥155 cm), the children from the shortest mothers (>145 cm) had about twice the risk of stunting (RR = 2.10; 95% CI: 1.97, 2.23) (Table 3). Mothers with short stature had about three times the risk of severe stunting (RR = 2.97; 95% CI: 2.65, 3.33) compared to tall mothers (Appendix A).

### 3.4. Interaction between Maternal Stature and Household Wealth and Its Effect on Stunting

When we tested the interaction between maternal height and household wealth in the model that measured the association between maternal height and offspring stunting adjusted for other covariates, we a found a robust significant interaction between maternal height categories and household wealth (*p* < 0.001) (Appendix A). From this adjusted model, we measured the combined effect of household wealth and maternal height on the prevalence of stunting (Figure 1). We found that the prevalence of stunting gradually increased from the wealthiest to the poorest wealth quintile among the children of the tall stature mothers (≥155 cm). However, for the children of the short stature mothers (<145 cm), the prevalence of stunting remained almost the same across the wealth quintiles from the wealthiest to the poorest.

### 3.5. Association of Maternal Stature with Offspring Wasting and Severe Wasting

We also found a robust significant association between maternal height and wasting after adjusting for covariates (Table 4, Appendix A). The adjusted models showed that a one-centimeter increase in maternal height was associated with a significant reduction in relative risk for child wasting (RR = 0.986; 95% CI: 0.980, 0.992) and severe wasting (RR = 0.984; 95% CI: 0.971, 0.997). Children of the short statured mothers (<145 cm) were significantly more likely to suffer from both wasting (RR = 1.28; 95% CI: 1.14, 1.43) and severe wasting (RR = 1.43; 95% CI: 1.11, 1.83) compared to tall mothers (≥155 cm).

## 4. Discussion

### 4.1. Main Findings

Based on our analysis of a nationally representative sample, maternal height was inversely associated with stunting and wasting among children under-five in Bangladesh. Our results provided robust evidence that children of mothers of short stature have a substantially higher risk of stunting and modest risk of wasting compared to the children of taller mothers. The finding that the intergenerational effect of undernutrition was not ameliorated by current household wealth after adjustment for other maternal and child level factors is an important finding. In Bangladesh and other LMICs where there is a high level of child undernutrition, the intergenerational effects of maternal undernutrition may lag the effects of interventions aimed at reducing child stunting. Monitoring maternal height may give a good indication of the reducing effects of intergenerational malnutrition.

### 4.2. Strength and Limitations

The strength of this study was that we used a nationally representative sample and pooled several rounds of surveys, which gave us a large enough sample size to estimate the effect [30]. Another strength was that we used the Poisson regression model with a robust variance that was suggested as one of the less biased approaches to obtain the correct estimates of the risk ratio for a dichotomous outcome like stunting or wasting [31]. It is a crucial study for Bangladesh where the high prevalence of child stunting and wasting remain as public health challenges despite the government and non-government policies and programs trying to improve maternal and child nutrition. Our study specifically assesses the intergenerational effect of maternal height on under-five child stunting and wasting after adjusting for well-established determinants of maternal and child undernutrition, such as age, birth history, social determinants like education, household wealth, residence, and location. However, our limitation was that we did not control previously established immediate causes of child malnutrition, such as inappropriate infant and young child feeding practices, household food insecurity, infectious disease, and poor health-seeking behavior in our analysis [32]. Although genetic or epigenetic factors are the intergenerational determinants jointly related to maternal and offspring nutrition, we cannot assess the effect of these factors because these data were not available in the BDHS. Moreover, it was beyond our scope to describe the mechanism of how maternal height and child growth faltering were associated, as we only tested the hypothesis that maternal height and child anthropometric outcome were associated in Bangladesh.

### 4.3. Association of Maternal Stature with Offspring Stunting and Wasting

Our findings of a robust inverse association between maternal height and stunting and wasting among children under-five were similar to other studies [12,13]. A large national survey (NFHS, 2005–2006) in India reported that each one-centimeter increase of maternal height was inversely associated with under-five stunting (RR = 0.971; 95% CI: 0.968, 0.0973) and wasting (RR = 0.989; 95% CI: 0.984, 0.994) [13]. Similarly, another large study that pooled data from 54 low-income countries showed that for every one-centimeter increase of maternal height significantly reduced the risk of stunting (RR = 0.968; 95% CI: 0.967, 0.968) and wasting RR = 0.994; 95% CI: 0.993, 0.995) among children under-five [12]. Even though this multi-country pooled analysis included BDHS 1997–2007, the present study analyzed the most recent nationally representative surveys for Bangladesh (BDHS 2004–2014). The findings from this study confirmed the intergenerational association between maternal stature and child undernutrition in Bangladesh.

### 4.4. Maternal Short Stature and the Risk of Offspring Stunting and Wasting

This study revealed a robust association between short stature of the mothers and the risk of their offspring stunting after adjustment for socioeconomic status, while this risk was greatest for the mothers whose stature was <145 cm. These findings have been similar to the results of previous studies [12,13]. The findings from other studies that investigated the intergenerational and other pathways of growth faltering can help explain the association between short maternal height and the risk of stunting. The intergenerational linkage between maternal short stature and the offspring’s growth faltering in utero can be explained by biomechanical (i.e., maternal organ size) and biological mechanisms (maternal nutrition stock, and fetal programming). Prior research has shown that short women are more likely to have narrower pelves which affect the uterine environment for optimum fetal growth and leads to the birth of low birth weight (LBW) babies [33].

Moreover, maternal short stature is an indicator of her cumulative net nutrition and biological deprivation over periods of rapid growth [34]. Poor nutritional status of women in pregnancy adversely affects placental growth that causes inadequate nutrient transfer and oxidative stress to the fetus. Nutrient deficiencies in utero cause epigenetic modification (i.e., DNA methylation) to alter fetal programming that results in fetal growth faltering, and delivery of LBW or SGA babies [35]. LBW or SGA infants born with nutrient deficiencies and immature immune systems are more susceptible to infection, while infection increases the risk of acute undernutrition by mucosal damage, impaired absorption of essential nutrients and loss of weight [36].

Along with exposure to intergenerational factors and infectious diseases, dietary factors like insufficient quality of complementary feeding have a paramount role in growth faltering in the first two years of life [37]. Poor diet quality with low dietary diversity is a predictor of micronutrient inadequacies [38], while the synergistic interaction between micronutrient deficiencies and infections causes growth faltering among children under-five in LMICs [39,40]. Evidence also suggests that household food insecurity is a predictor of child undernutrition and linear growth faltering in diverse settings of LMICs [41,42]. Acute growth faltering is reflected as wasting in infancy and childhood, which can lead to long-term linear growth faltering or stunting [43], if not treated with immediate inputs such as appropriate infant feeding, dietary diversity, disease prevention, or health facilities and underlying causes like household food security. Hence, the intergenerational effects of growth impairment in utero interact with other immediate and underlying determinants of undernutrition influencing growth faltering from birth to five years of age.

### 4.5. Interactions between Household Wealth and Maternal Short Stature on Child Stunting

This study showed an important interaction between maternal stature and household wealth and its association with child stunting. The interaction showed that amongst short mothers there was a similarly high prevalence of stunting across all wealth quintiles from the wealthiest to the poorest families. However, this study found an effect of social inequalities on stunting, which has been reported in earlier studies [44,45], among the children of the tall mothers but not among the children of the short statured mothers. These findings demonstrate that the intergenerational effect of undernutrition remains important beyond the current wealth status of the household. Moreover, these findings are consistent with the recent concept of “the stunting syndrome” that proposes an intergenerational transmission of poor nutrition from mother to child, in which, short women, who were stunted in childhood, are more likely to have stunted children and thus, generating an intergenerational cycle of poverty [46]. These findings suggest that without addressing intergenerational factors, immediate determinants of food, care, health, or economic input will not be sufficient to overcome the long-term negative impact of intergenerational transmission of poor nutrition on child linear growth.

### 4.6. Policy Implications

In Bangladesh, a major contributor to social disparities in child stunting is maternal short stature [45]. The country has the highest proportion of women in the world with short stature (height < 145 cm), with more than one in ten women being short in stature or stunted (height < 145 cm) [47]. The high burden of maternal short stature implies that there has been little improvement in attaining adequate nutrition for women in adulthood for optimum fetal and child nutrition. A report from the MINIMat trial, which targeted pregnant women in rural Bangladesh, showed that combined food and micronutrient supplements in pregnancy were not effective in reducing child growth failure, where nearly one-third of the newborns had low birth weight [48]. This study also assumed that the high proportion of LBW might be related to the average short stature of the mothers (149.8 cm) who participated in this trial.

Furthermore, in Bangladesh, where inappropriate infant and young child feeding, poor sanitation, and infectious diseases are highly prevalent, LBW children who are born with compromised immunity are at high risk of infections, wasting, and stunting [36,49]. Evidence suggests that one-fourth of child growth faltering occurs in utero and continues until two years of age, and then slowly continues until five years of age in LMICs, including Bangladesh [50]. Therefore, a life-cycle approach is needed for nutrition-health related interventions for young girls, adolescents, preconception, and the first thousand days of life, which are ‘critical windows of opportunity’ for child nutrition [51,52]. Recently, United Nations System Standing Committee on Nutrition recommended improving maternal nutrition, with a special emphasis on short stature women, through improving preconception or conception diet quality to ensure optimum nutrition in utero to break the intergenerational cycle of growth faltering of the fetus that leads to LBW and stunting at a later stage of life [53]. Evidence also suggests that nutrition interventions such as promotion of nutrition and improved water and sanitation practices can improve nutrition and growth to break the vicious cycle of growth faltering in childhood [46]. As nutrition interventions have long term consequences on adult height, they should also target girls, especially adolescent girls, to ensure they get adequate nutrition to achieve the optimum height in adulthood to minimize short stature. Hence, the current study findings draw attention to program managers and policymakers to focus on improving nutrition throughout the lifecycle to reduce the risk of intergenerational transmission of undernutrition from mothers to their child to prevent stunting as well as wasting in the countries like Bangladesh.

## 5. Conclusions

In summary, short maternal height is associated with an increased risk of stunting and wasting among children under-five in Bangladesh. This finding suggests an intergenerational linkage between maternal and child chronic undernutrition that will need addressing for sustained improvements in maternal and child nutrition to reduce under-five stunting in the current context of Bangladesh.

## Figures and Tables

**Figure 1 nutrients-11-01818-f001:**
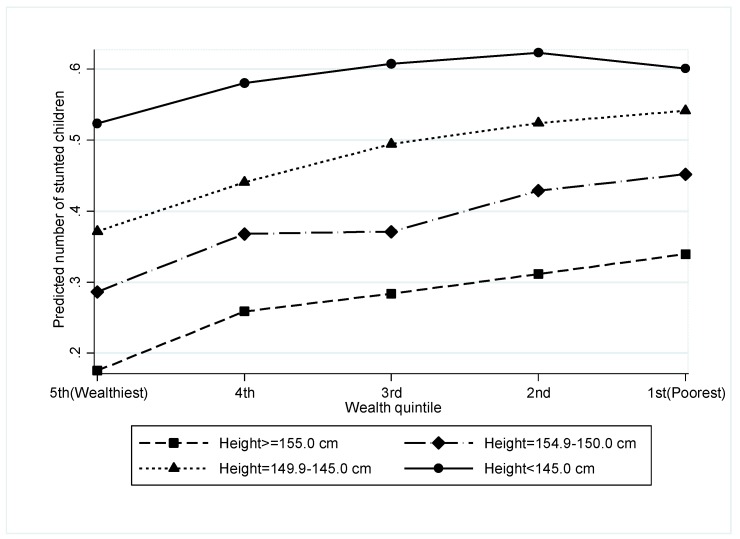
Combined effect of household wealth and maternal height on stunting adjusted for other covariates.

**Table 1 nutrients-11-01818-t001:** Percentage of the children under-five who were stunted, wasted, severely stunted, and severely wasted and their maternal, birth, and socio-demographic characteristics.

Covariates	All Livebirths, N = 25,635									
	Stunted *		Wasted ^+^		Severely Stunted **		Severely Wasted ^++^		Total	
	n	%	n	%	n	%	n	%	n	%
Maternal covariates										
Maternal Height (cm)categories										
≥155.0 cm (tall)	1444	13.5	770	19.1	438	10.8	161	18.0	5634	21.6
154.9–150.0 cm	3294	30.6	1300	33.1	1087	26.5	292	33.9	8746	33.9
149.9–145.0 cm	3807	35.6	1203	31.3	1536	37.3	261	31.4	7778	30.7
<145.0 cm (short)	2156	20.4	629	16.5	1030	25.5	152	16.6	3477	13.8
Age at birth, y										
<20	3111	29.4	1092	29.4	1175	29.1	233	28.1	7085	28.3
20–24	3539	33.3	1300	32.5	1297	31.9	264	29.4	8619	33.5
25–29	2192	20.5	844	22.0	834	20.3	204	24.7	5690	22.1
≥30	1859	16.9	666	16.1	785	18.8	165	17.8	4241	16.0
Educational level										
No education	3241	30.4	1049	27.3	1477	36.2	232	26.9	6027	24.4
Primary	3690	34.5	1292	32.8	1480	35.9	295	32.6	7740	30.3
Secondary	3368	31.7	1326	34.5	1037	25.6	285	35.0	9782	38.2
Higher	402	3.4	235	5.4	97	2.3	54	5.4	2086	7.0
Occupation										
Not working	8607	79.2	3121	78.7	3310	79.6	717	82.3	20,914	80.7
Working	2094	20.8	781	21.3	781	20.4	149	17.7	4721	19.3
Child Covariates										
child age category, mo										
0–11	1044	9.7	800	21.3	333	8.2	237	28.6	5001	19.7
12–23	2310	21.9	913	23.3	893	22.1	214	24.9	5121	20.1
24–35	2489	22.9	737	18.5	998	23.6	171	17.2	5108	19.6
36–47	2616	24.4	701	17.9	1047	25.9	129	15.6	5261	20.5
48–59	2242	21.2	751	19.1	820	20.3	115	13.8	5144	20.2
Birth Order										
First	3376	31.4	1325	33.6	1169	28.4	286	32.5	9009	34.9
Second	2708	25.9	1005	26.4	967	24.4	203	24.2	6923	27.1
Third	1845	17.4	681	17.5	717	17.3	153	17.7	4364	17.4
Fourth	1180	11.0	419	10.8	496	12.4	105	12.2	2453	9.7
≥Fifth	1592	14.4	472	11.8	742	17.6	119	13.5	2886	11.0
Birth Interval										
First child	3376	31.4	1325	33.6	1169	28.4	286	32.5	9009	34.9
≤23 months	1123	10.0	329	8.3	520	11.6	72	8.3	2155	8.1
24–47 months	3228	30.4	1047	27.8	1377	34.3	243	29.3	6546	26.0
≥48 months	2974	28.2	1201	30.3	1025	25.7	265	30.0	7925	31.0
Sex of the child										
Male	5487	50.9	2085	53.1	2127	51.7	488	56.6	13,060	50.9
Female	5214	49.1	1818	46.9	1964	48.3	378	43.4	12,575	49.1
Household covariates										
Wealth Quintile										
First, poorest	2965	25.0	961	22.1	1288	29.2	225	24.1	5653	19.8
Second	2414	24.1	814	22.3	1023	26.3	166	20.3	4722	20.0
Third	2152	20.8	782	20.7	823	20.9	177	20.5	4900	19.9
Fourth	1833	17.9	727	18.7	588	15.1	160	18.3	5011	20.0
Fifth, richest	1337	12.3	618	16.2	369	8.5	138	16.8	5349	20.2
Father’s Education										
No education	4136	39.4	1322	34.4	1798	44.6	303	35.5	7865	32.2
Primary	3372	31.1	1170	29.6	1308	31.1	264	31.1	7363	28.7
Secondary	2465	23.1	1034	26.5	800	20.1	217	25.3	7100	27.4
Higher	728	6.5	376	9.5	185	4.2	82	8.2	3307	11.7
Location of Residence										
Urban	2924	19.0	1083	19.3	1051	17.9	251	21.1	8068	22.1
Rural	7777	81.0	2819	80.7	3040	82.1	615	78.9	17,567	77.9
Region										
Barisal	1322	6.4	446	5.8	522	6.9	89	5.3	2979	5.8
Chittagong	2253	22.7	802	23.0	934	24.5	185	24.8	5181	21.9
Dhaka	2061	32.5	668	29.1	770	32.0	150	28.3	4855	32.3
Khulna	1023	7.6	467	9.6	297	5.9	107	10.0	3052	9.3
Rajshahi	1396	15.1	626	17.9	462	13.8	135	16.3	3739	16.1
Sylhet	1526	9.8	539	9.8	628	10.3	121	10.7	3513	9.6
Rangpur	1120	5.8	354	4.8	478	6.6	79	4.7	2316	5.0
Year of survey										
2004	2903	27.3	847	22.1	1232	31.3	286	32.5	9009	34.9
2007	2229	20.7	876	22.4	871	21.0	72	8.3	2155	8.1
2011	3041	28.3	1175	29.7	1159	27.9	243	29.3	6546	26.0
2014	2528	23.6	1004	25.9	829	19.8	265	30.0	7925	31.0
Total	10,701	42.1	3902	15.4	4091	15.8	866	3.4	25,635	100

Note: * Stunted: height-for-age Z-score < −2 SD ** Severely stunted: height-for-age Z-score < −3 SD ^+^ Wasted: weight-for-height Z-score < −2 SD ^++^ Severely wasted: weight-for-height Z-score < −3 SD.

**Table 2 nutrients-11-01818-t002:** Mean maternal height in cm (95% confidence intervals) among the children under-five who were or were not stunted, wasted, severely stunted, and severely wasted.

Anthropometric Category		N	Mean in cm (95% CI)	Mean Difference (95% CI)	*p* Value
Stunted *	No	14,934	151.9 (151.8, 152.0)	2.6 (2.5, 2.8)	<0.001
	Yes	10,701	149.3 (149.2, 149.4)		
Severely stunted **	No	21,544	151.3 (151.2, 151.3)	2.8 (2.6, 3.0)	<0.001
	Yes	4091	148.4 (148.3, 148.6)		
Wasted ^+^	No	21,734	150.9 (150.8, 151.0)	0.6 (0.4, 0.8)	<0.001
	Yes	3901	150.3 (150.2, 150.5)		
Severely wasted ^++^	No	24,769	150.8 (150.8, 150.9)	0.6 (0.2, 0.9)	0.003
	Yes	866	150.3 (149.9, 150.7)		

Note: * Stunted: height-for-age Z-score < −2 SD ** Severely stunted: height-for-age Z-score < −3 SD ^+^ Wasted: weight-for-height Z-score < −2 SD ^++^ Severely wasted: weight-for-height Z-score < −3 SD.

**Table 3 nutrients-11-01818-t003:** Association of maternal height (cm) with stunted children under-five adjusted for maternal and other covariates showing unadjusted and adjusted relative risk with 95% confidence intervals.

Covariates	Stunted * Under-Five Children					
	Unadjusted		Adjusted Model 1 ^a^		Adjusted Model 2 ^b^	
Maternal Covariates	RR (95% CI)	*p* Value	RR (95% CI)	*p* Value	RR (95% CI)	*p* Value
Maternal height per 1-cm increase	0.954 (0.951, 0.958)	<0.001	0.960 (0.957, 0.963)	0.001		
Maternal height (cm) categories						
≥155.0 cm (tall)	1 (Reference)				1 (Reference)	
154.9–150.0 cm	1.45 (1.36, 1.54)				1.40 (1.32, 1.48)	
149.9–145.0 cm	1.86 (1.75, 1.98)				1.74 (1.64, 1.84)	
<145.0 cm (short)	2.36 (1.22, 2.54)	<0.001			2.10 (1.97, 2.23)	<0.001
Maternal Age at birth, y						
<20	1 (Reference)		1 (Reference)		1 (Reference)	
20–24	0.96 (0.92, 1.00)		0.93 (0.89, 0.98)		0.93 (0.89, 0.98)	
25–29	0.89 (0.85, 0.94)		0.82 (0.77, 0.87)		0.82 (0.77, 0.87)	
≥30	1.02 (0.97, 1.07)	<0.001	0.81 (0.75, 0.87)	<0.001	0.81 (0.76, 0.88)	<0.001
Maternal Educational level						
No education	1 (Reference)		1 (Reference)		1 (Reference)	
Primary	0.91 (0.88, 0.95)		1.02 (0.98, 1.06)		1.02 (0.98, 1.06)	
Secondary	0.67 (0.64, 0.70)		0.94 (0.89, 0.99)		0.94 (0.89, 0.99)	
Higher	0.39 (0.35, 0.44)	<0.001	0.81 (0.72, 0.92)	<0.001	0.81 (0.72, 0.92)	<0.001
Mother’s Occupation						
Not working	1 (Reference)					
Working	1.09 (1.05, 1.14)	<0.001				
**Child Covariates**						
Child age category, mo						
0–11	1 (Reference)		1 (Reference)		1 (Reference)	
12–23	2.21 (2.05, 2.38)		2.21 (2.05, 2.37)		2.20 (2.05, 2.37)	
24–35	2.37 (2.20, 2.55)		2.34 (2.18, 2.52)		2.34 (2.18, 2.51)	
36–47	2.42 (2.25, 2.60)		2.36 (2.20, 2.54)		2.36 (2.20, 2.53)	
48–59	2.12 (1.97, 2.29)	<0.001	2.03 (1.89, 2.18)	<0.001	2.04 (1.89, 2.19)	<0.001
Birth Order						
First	1 (Reference)		1 (Reference)		1 (Reference)	
Second	1.06 (1.01, 1.11)		1.06 (1.01, 1.11)		1.05 (1.00, 1.11)	
Third	1.12 (1.06, 1.18)		1.08 (1.01, 1.15)		1.08 (1.01, 1.15)	
Fourth	1.27 (1.20, 1.34)		1.20 (1.11, 1.29)		1.19 (1.11, 1.28)	
≥Fifth	1.46 (1.39, 1.54)	<0.001	1.30 (1.20, 1.40)	<0.001	1.29 (1.19, 1.39)	<0.001
Birth Interval						
First child	1 (Reference)					
≤23 months	1.37 (1.29, 1.45)					
24–47 months	1.30 (1.25, 1.36)					
≥48 months	1.10 (0.97, 1.06)	<0.001				
Sex of the child						
Male	1 (Reference)					
Female	1.00 (9.67, 1.04)	0.999				
Household covariates						
Wealth Quintile						
Fifth (wealthiest)	1 (Reference)		1 (Reference)		1 (Reference)	
Fourth	1.47 (1.37, 1.58)		1.26 (1.18, 1.35)		1.25 (1.17, 1.34)	
Third	1.72 (1.62, 1.84)		1.34 (1.26, 1.44)		1.34 (1.25, 1.43)	
Second	1.99 (1.87, 2.11)		1.46 (1.37, 1.56)		1.45 (1.36, 1.55)	
First (poorest)	2.09 (1.97, 2.22)	<0.001	1.51 (1.41, 1.61)	<0.001	1.50 (1.40, 1.60)	<0.001
Father’s Education						
No education	1 (Reference)		1 (Reference)		1 (Reference)	
Primary	0.88 (0.85, 0.92)		0.98 (0.94, 1.02)		0.98 (0.94, 1.02)	
Secondary	0.69 (0.66, 0.72)		0.90 (0.86, 0.95)		0.90 (0.85, 0.94)	
Higher	0.45 (0.42, 0.49)	<0.001	0.76 (0.69, 0.83)	<0.001	0.75 (0.69, 0.82)	<0.001
Location of Residence						
Urban	1 (Reference)		1 (Reference)		1 (Reference)	
Rural	1.22 (1.17, 1.27)	<0.001	1.08 (1.04, 1.13)	<0.001	1.08 (1.04, 1.12)	<0.001
Region						
Barisal	1 (Reference)		1 (Reference)		1 (Reference)	
Chittagong	0.95 (0.89, 1.00)		1.01 (0.96, 1.06)		1.02 (0.97, 1.07)	
Dhaka	0.92 (0.87, 0.97)		0.95 (0.90, 1.01)		0.96 (0.91, 1.01)	
Khulna	0.75 (0.70, 0.80)		0.84 (0.79, 0.89)		0.84 (0.79, 0.90)	
Rajshahi	0.86 (0.80, 0.91)		0.83 (0.78, 0.88)		0.83 (0.78, 0.88)	
Sylhet	0.93 (0.88, 0.99)		0.93 (0.88, 0.99)		0.94 (0.88, 0.99)	
Rangpur	1.07 (1.00, 1.14)	<0.001	1.15 (1.07, 1.22)	<0.001	1.15 (1.08, 1.23)	<0.001
Year of survey						
2004	1 (Reference)		1 (Reference)		1 (Reference)	
2007	0.89 (0.85, 0.93)		0.93 (0.90, 0.98)		0.94 (0.90, 0.98)	
2011	0.81 (0.78, 0.85)		0.88 (0.84, 0.92)		0.88 (0.85, 0.92)	
2014	0.72 (0.69, 0.76)		0.82 (0.78, 0.89)	<0.001	0.83 (0.78, 0.87)	<0.001
Recall	1.003 (1.002, 1.003)					

Note: Abbreviations: * Stunted: height-for-age Z-score < −2 SD; RR: Relative Risk; CI: Confidence Interval; ^a^ Adjusted model 1 estimated the association between stunting and maternal height (cm) adjusted for other covariates, in which maternal height (cm) was considered as the continuous variable; ^b^ Adjusted model 2 estimated the association of stunting with maternal height (cm) catagories, ≥155.0 cm (tall, reference group), 154.9–150.0 cm, 149.9–145.0 cm, and <145.0 cm (short), adjusted for other.

**Table 4 nutrients-11-01818-t004:** Association of maternal height (cm) with wasted children under-five adjusted for other covariates showing unadjusted and adjusted relative risk with 95% confidence intervals.

Covariates	Wasted ^+^ Under-Five Children					
	Unadjusted		Adjusted Model 1 ^a^		Adjusted Model 2 ^b^	
	RR (95% CI)	*p* Value	RR (95% CI)	*p* Value	RR (95% CI)	*p* Value
**Maternal Covariates**						
Maternal height per 1-cm increase	0.983 (0.977, 0.989)	<0.001	0.986(0.980, 0.992)	<0.001		
Maternal height (cm) categories						
≥155.0 cm (tall)	1 (Reference)				1 (Reference)	
154.9–150.0 cm	1.11 (1.01, 1.22)				1.09 (0.99, 1.21)	
149.9–145.0 cm	1.16 (1.05, 1.28)				1.13 (1.02, 1.25)	
<145.0 cm (short)	1.35 (1.21, 1.51)	<0.001			1.28 (1.14, 1.43)	<0.001
Maternal Age at birth, y						
<20	1 (Reference)					
20–24	0.94 (0.86, 1.02)					
25–29	0.96 (0.87, 1.06)					
≥30	0.97 (0.88, 1.08)	0.487				
Maternal Educational level						
No education	1 (Reference)		1 (Reference)		1 (Reference)	
Primary	0.97 (0.89, 1.06)		0.98 (0.90, 1.07)		0.98 (0.90, 1.07)	
Secondary	0.81 (0.74, 0.88)		0.85 (0.77, 0.94)		0.85 (0.77, 0.94)	
Higher	0.69 (0.59, 0.81)	<0.001	0.80 (0.67, 0.95)	0.002	0.80 (0.67, 0.95)	0.002
Mother’s Occupation						
Not working	1 (Reference)		1 (Reference)		1 (Reference)	
Working	1.13 (1.04, 1.23)	0.005	1.13 (1.04, 1.24)	0.006	1.13 (1.04, 1.24)	0.007
**Child Covariates**						
Child age category, mo						
0–11	1 (Reference)		1 (Reference)		1 (Reference)	
12–23	1.07 (0.97, 1.19)		1.07 (0.97, 1.18)		1.07 (0.97, 1.18)	
24–35	0.87 (0.78, 0.97)		0.86 (0.77, 0.96)		0.86 (0.77, 0.96)	
36–47	0.81 (0.73, 0.90)		0.79 (0.71, 0.88)		0.79 (0.71, 0.88)	
48–59	0.87 (0.79, 0.97)	<0.001	0.84 (0.76, 0.94)	<0.001	0.85 (0.76, 0.94)	<0.001
Birth Order						
First	1 (Reference)					
Second	1.01 (0.92, 1.11)					
Third	1.05 (0.95, 1.15)					
Fourth	1.16 (1.03, 1.30)					
≥Fifth	1.12 (1.00, 1.25)	0.059				
Birth Interval						
First child	1 (Reference)					
≤23 months	1.06 (0.93, 1.20)					
24–47 months	1.11 (1.02, 1.22)					
≥48 months	1.02 (0.94, 1.11)	0.094				
Sex of the child						
Male						
Female	0.92 (0.86, 0.98)	0.012	0.91 (0.85, 0.98)	0.009	0.91 (0.85, 0.98)	0.009
Household covariates						
Wealth Quintile						
First, poorest	1 (Reference)		1 (Reference)		1 (Reference)	
Second	0.99 (0.90, 1.10)		0.97 (0.87, 1.08)		0.97 (0.87, 1.08)	
Third	0.93 (0.84, 1.02)		0.92 (0.83, 1.02)		0.92 (0.83, 1.02)	
Fourth	0.84 (0.75, 0.93)		0.87 (0.78, 0.98)		0.87 (0.78, 0.98)	
Fifth, richest	0.72 (0.64, 0.80)	<0.001	0.82 (0.72, 0.93)	0.016	0.82 (0.72, 0.93)	0.017
Father’s Education						
No education	1 (Reference)					
Primary	0.96 (0.89, 1.05)					
Secondary	0.91 (0.83, 0.99)					
Higher	0.76 (0.67, 0.87)	0.001				
Location of Residence						
Urban	1 (Reference)		1 (Reference)		1 (Reference)	
Rural	1.19 (1.10, 1.28)	<0.001	1.12 (1.03, 1.22)	0.009	1.12 (1.03, 1.22)	0.010
Region						
Barisal	1 (Reference)		1 (Reference)		1 (Reference)	
Chittagong	1.06 (0.95, 1.19)		1.10 (0.98, 1.23)		1.10 (0.98, 1.24)	
Dhaka	0.91 (0.81, 1.03)		0.92 (0.82, 1.04)		0.93 (0.82, 1.04)	
Khulna	1.04 (0.91, 1.18)		1.10 (0.96, 1.25)		1.10 (0.96, 1.25)	
Rajshahi	1.13 (0.99, 1.28)		1.09 (0.97, 1.24)		1.09 (0.97, 1.24)	
Sylhet	1.04 (0.91, 1.19)		1.03 (0.90, 1.17)		1.03 (0.90, 1.18)	
Rangpur	0.99 (0.85, 1.14)	0.010	0.99 (0.85, 1.15)	0.009	0.99 (0.85, 1.15)	0.009
Year of survey						
2004	1 (Reference)		1 (Reference)		1 (Reference)	
2007	1.18 (1.07, 1.30)		1.21 (1.10, 1.33)		1.21 (1.10, 1.34)	
2011	1.05 (0.96, 1.15)		1.15 (1.05, 1.27)		1.16 (1.05, 1.27)	
2014	0.98 (0.88, 1.08)	0.001	1.08 (0.97, 1.20)	0.001	1.08 (0.97, 1.20)	0.001
Recall	0.999 (0.998, 1.000)	0.163				

Note: Abbreviations: ^+^ Wasted: weight-for-height Z-score < −2 SD; RR: Relative Risk; CI: Confidence Interval; ^a^ Adjusted model 1 estimated the association between wasting and maternal height (cm) adjusted for other covariates, in which maternal height (cm) was considered as the continuous variable; ^b^ Adjusted model 2 estimated the association of wasting with maternal height (cm) categories, ≥155.0 cm (tall, reference group), 154.9–150.0 cm, 149.9–145.0 cm, and <145.0 cm (short), adjusted for other covariates.

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
