# Peer review of "Assessing the Intergenerational Linkage between Short Maternal Stature and Under-Five Stunting and Wasting in Bangladesh"

_nutrients, 2019, doi:10.3390/nu11081818_

Round 1

Reviewer 1 Report

The manuscript by Khatun et al sought to assess the role of maternal short stature on offspring stunting and wasting in Bangladesh at a national level.  The authors used pooled data for a sample size of 28, 123 mother-infant pairs.  This study had many strengths that included a large sample size, controlling for multiple co-variates, appropriate statistical analyses, and good organization and flow.  The authors found that every 1 cm increase in maternal height was associated with significantly decreased risk for childhood stunting and wasting.  Interestingly, prevalence of stunting increased as wealth declined, but only in the children born to the tallest mothers.  This relationship was not present in the children born to the shortest mothers.  However, despite these strengths, there were concerns and questions. 

Concerns and Questions:

1.     Why was birth order used as a child factor in the analysis as opposed to using parity as a maternal factor?

2.     The last sentence of the “Conclusion” is overstated.  These findings do not provide evidence of an intergenerational pathway as much as using maternal height as an indicator for childhood stunting and wasting risk.    

3.     Why was ethnicity not considered as a factor?

4.     Was maternal smoking data collected? 

5.     Infant sex was a co-variate, but were there sex differences?  Stratifying by sex may be very interesting.

6.     The children born to the shortest mothers may have a stronger genetic risk for stunting and wasting, or conversely, those in the tallest group may be more apt to be tall due to underlying genetics.  Can the authors please comment?

7.     On Line 264-266, what do the authors mean by “data constraints”?

8.     In section 4.5, the authors state the association between maternal height and wasting is “robust”, but the relative risk did not look robust (0.983 for wasting, and 0.984 for severe wasting) despite being statistically significant. 

9.     On Line 295-297 doesn’t make sense as written. Do the authors mean inactivating gene expression in pathways involved in growth hormone production, metabolic disorders, etc?

10.  On Line 175 change to “maternal”   

11.  On Line 200 change to “association”

Author Response

Response to Reviewer 1 Comments

We would like to thank the reviewer 1 for his valuable and constructive comments that helped me to improve the manuscripts. We have addressed the comments point by point. We have also revised the manuscript based on the comments. Our responses to the reviewer’s comments are as below:

Point 1. Why was birth order used as a child factor in the analysis as opposed to using parity as a maternal factor?

Response 1: We agree with the reviewer that parity is a potentially important maternal factor  associated with stunting. However, we have chosen the birth order to be consistent with similar previous studies that assessed the association between maternal height and child anthropometry.

Point 2: The last sentence of the “Conclusion” is overstated.  These findings do not provide evidence of an intergenerational pathway as much as using maternal height as an indicator for childhood stunting and wasting risk. 

Response 2: We agree with the reviewer and changed the word ‘pathway’ to ‘linkage’. I have also revised the sentence as below:

“This finding suggests an intergenerational linkage between maternal and child chronic undernutrition that will need addressing for sustained improvements in maternal and child nutrition to reduce under-five stunting in the current context of Bangladesh.”

Point 3: Why was ethnicity not considered as a factor?

Response 3: The population of Bangladesh is ethnically homogenous. And the data for ethnicity is not available in the Bangladesh demographic and health survey. Therefore, we did not consider ethnicity in our analysis.

Point 4: Was maternal smoking data collected? 

Response 4 : No as maternal smoking data were not available. An earlier study reported female smoking prevalence in Bangladesh of 1.5% (Nargis N, Thompson ME, Fong GT, et al. Prevalence and Patterns of Tobacco Use in Bangladesh from 2009 to 2012: Evidence from International Tobacco Control (ITC) Study. PLoS One. 2015;10(11):e0141135. Published 2015 Nov 11. doi:10.1371/journal.pone.0141135)

Point 5: Infant sex was a co-variate, but were there sex differences?  Stratifying by sex may be very interesting.

Response 5: Thanks to the reviewer for his suggestion. There were no significant sex differences for stunting or for wasting.

As suggested, we have done the analysis stratifying by sex. However, the stratified findings were almost similar to the results without stratification. Therefore, we decided not to present the results stratified by sex.

Point 6: The children born to the shortest mothers may have a stronger genetic risk for stunting and wasting, or conversely, those in the tallest group may be more apt to be tall due to underlying genetics.  Can the authors please comment?

Response 6: We agree with the reviewer. However, despite the genetic risk, the children of the short mothers are at high risk for growth retardation due to the intergenerational transmission of poor nutrition from the mother to the child in utero. The short stature of the woman reflects her chronic deprivation of optimum nutrition and living standards over the growing period throughout the life cycle (from birth to childhood to adolescence). Short women who have poor health stock and may not transmit the essential nutrients to the fetus during pregnancy. The environmental factors like food insecurity, poor diet quality, low-quality water, and sanitation may also influence the nutrient deficiencies among the women, either tall or short, during pregnancy. In pregnancy, deficiencies of methyl donor nutrients such as choline, vitamin B6, B12, zinc, betaine and folic acid adversely affects epigenetic modification and foetal metabolic programming that causes intra-uterine growth retardation.  New research in early human development shows that epigenetic adaptations to the early life environment occur from conception and that these adaptations affect development and adverse health outcome throughout the life course. Hence, both genetic and environmental risk needs to be considered.

Point 7.     On Line 264-266, what do the authors mean by “data constraints”?

Response 7: Here, we mean ‘data constraints’ as the limitation of the data in the Bangladesh Demographic and Health Survey (BDHS). Data on genetic and epigenetic factors were not available in the BDHS.  We have revised the sentence as follows:

“Although genetic or epigenetic factors are the intergenerational determinants jointly related to maternal and offspring nutrition, we cannot assess the confounding effect of these factors because these data were not available in the BDHS.”

Point 8.     In section 4.5, the authors state the association between maternal height and wasting is “robust”, but the relative risk did not look robust (0.983 for wasting, and 0.984 for severe wasting) despite being statistically significant. 

Response 8: We agree with the reviewer, but we state the ‘robust’ here as it has been found that there was a very strong association between maternal height and wasting (p<0.001).

Point 9.     On Line 295-297 doesn’t make sense as written. Do the authors mean inactivating gene expression in pathways involved in growth hormone production, metabolic disorders, etc?

Response 9: We have revised the sentence. Please see below:

"Nutrient deficiencies in utero cause epigenetic modification (i.e., DNA methylation) to alter fetal programming including inactive gene expression of growth hormone, metabolic disorder, organ dysfunction, and defects in cell signaling that results in fetal growth faltering, and delivery of LBW or SGA babies."

Point 10.  On Line 175 change to “maternal”  

Response 10: It has been corrected as advised. 

Point 11.  On Line 200 change to “association”

Response 11: It has been corrected as suggested.

Reviewer 2 Report

Dear Authors,

    I have enjoyed the reading of your manuscript entitled ““Assessing the intergenerational linkage between short maternal stature and under-five stunting and wasting in Bangladesh” submitted to Nutrients.

Although the association between maternal stature and stunting and wasting in the progeny is well-known, your work provides us with a solid and well-argued evidence from the analysis of representative samples of Bangladesh at four different time points.

The statistical analysis is very valuable, with an exhaustive consideration of potential covariates at individual, family and community levels. The Poison Regression offers a proper approach and limits the risk of bias.

I have found very interesting points in your work, such as the variable “recall period” (not included in many studies), clear tables and figures and a very remarkable discussion.

Congratulations for such an amazing work.

Author Response

We would like to thank the reviewer for his thoughtful and encouraging  comments.